# 3D Bioprinting of Human Tissues: Biofabrication, Bioinks, and Bioreactors

**DOI:** 10.3390/ijms22083971

**Published:** 2021-04-12

**Authors:** Jianhua Zhang, Esther Wehrle, Marina Rubert, Ralph Müller

**Affiliations:** Institute for Biomechanics, ETH Zurich, Leopold-Ruzicka-Weg 4, 8093 Zurich, Switzerland; jianhua.zhang@hest.ethz.ch (J.Z.); esther.wehrle@hest.ethz.ch (E.W.); marina.rubert@hest.ethz.ch (M.R.)

**Keywords:** tissue engineering, 3D bioprinting, bioink, bioreactor

## Abstract

The field of tissue engineering has progressed tremendously over the past few decades in its ability to fabricate functional tissue substitutes for regenerative medicine and pharmaceutical research. Conventional scaffold-based approaches are limited in their capacity to produce constructs with the functionality and complexity of native tissue. Three-dimensional (3D) bioprinting offers exciting prospects for scaffolds fabrication, as it allows precise placement of cells, biochemical factors, and biomaterials in a layer-by-layer process. Compared with traditional scaffold fabrication approaches, 3D bioprinting is better to mimic the complex microstructures of biological tissues and accurately control the distribution of cells. Here, we describe recent technological advances in bio-fabrication focusing on 3D bioprinting processes for tissue engineering from data processing to bioprinting, mainly inkjet, laser, and extrusion-based technique. We then review the associated bioink formulation for 3D bioprinting of human tissues, including biomaterials, cells, and growth factors selection. The key bioink properties for successful bioprinting of human tissue were summarized. After bioprinting, the cells are generally devoid of any exposure to fluid mechanical cues, such as fluid shear stress, tension, and compression, which are crucial for tissue development and function in health and disease. The bioreactor can serve as a simulator to aid in the development of engineering human tissues from in vitro maturation of 3D cell-laden scaffolds. We then describe some of the most common bioreactors found in the engineering of several functional tissues, such as bone, cartilage, and cardiovascular applications. In the end, we conclude with a brief insight into present limitations and future developments on the application of 3D bioprinting and bioreactor systems for engineering human tissue.

## 1. Introduction

Tissue engineering is a multidisciplinary field that uses a combination of cells, biomaterials, and engineering technologies to develop artificial biological tissue substitutes [1,2]. The concept and scope have significantly expanded during the past decades, leading to two major areas: (i) developing new methods to repair, regenerate, and replace damaged tissues, and (ii) creating in vitro tissue models to better understand tissue development, disease mechanism, and to test and screen drugs [3,4,5]. Despite significant advancement in the field of tissue engineering, there is still a continuous shortage of tissues for transplantation or insufficient tissue regeneration. Besides, there is a lack of tissue models with complex multiscale architecture and tissue–tissue interfaces for drug discovery and testing [6]. The conventional tissue engineering approaches use three-dimensional (3D) prefabricated scaffolds as matrices to load cells [7,8]. The scaffolds serve as 3D templates that support cells to attach, proliferate, differentiate and secrete an extracellular matrix (ECM), which eventually leads to the generation of mature cell-laden grafts with comparable properties to their native counterparts. However, these conventional scaffold-based approaches are limited by the intrinsic inability to mimic the complex microstructures of biological tissues and are unable accurately to define the spatial location and distribution of cells [9,10].

3D bioprinting is an emerging technology expected to revolutionize the field of tissue engineering and regenerative medicine. As an additive manufacturing technique, 3D bioprinting shows promise for creating complex composite tissue constructs through precise placement of living cells and biomaterials in a layer-by-layer fashion [11,12]. The 3D bioprinters’ ability to deposit biomaterials with micrometer precision in cell-friendly conditions gives it an advantage over conventional scaffold-based approaches since it shows effective control over scaffold fabrication and cell distribution [13].

Figure 1 shows the typically followed steps for the production of 3D bioprinted human tissues, which includes pre-processing, processing and post-processing stages. The pre-processing step is to culture human cells and designs a scaffold model for 3D bioprinting. Human-specific cells were isolated and expanded to achieve a large number of cells in vitro. Magnetic resonance (MRI) or computed tomography (CT) imaging technologies are explored to acquire structure and morphology information of the targeting tissues. The recorded images are reconstructed to achieve the 3D bioprinting models and then transfer to the model file, which can be read by the bioprinter, such as gcode [14,15]. Some bioprinting companies also provide professional commercial software (e.g., Axway TradeSync Integration Manager^®^, BioAssemblyBot^®^, and BioCAD^®^) to design, draw, and print multiscale structures ranging from cells to tissue constructs.

The processing step is to fabricate the 3D cell-laden constructs by 3D bioprinting. The most commonly used bioprinting systems are based on three major strategies: inkjet, laser, and extrusion-based bioprinting [16]. In most cases, a three-axis mechanical platform controls the movements of extruders printing the biomaterials in the required algorithm and shape based on the 3D tissue models [17]. The biomaterial that is printed is referred to as a “bioink,” which can be defined as an ink formulation that allows the printing of living cells and growth factors. The selection of proper bioink is crucial for successful bioprinting. Because it will provide the required properties for adequate printing fidelity and mechanical properties to ensure printability and long-term functionality following deposition [12]. Post-processing involves the maturation of cell-laden constructs to reinforce the development of desired tissue constructs. The optimal nutrition and oxygen delivery, as well as removal of wastes, are required to maintain cell viability and functionality. More importantly, chemical and mechanical cues are of critical importance to direct cellular behaviors and human tissue development [18,19]. Growth and differentiation factors are commonly used and carefully chosen as chemical stimulation to drive specific cell responses. Growth factors play a major role in cell division, matrix synthesis, and tissue differentiation [20]. The need for in vitro mechanical stimulation in tissue engineering is drawn from the fact that most tissues function under specific biomechanical environments in vivo. These mechanical environments play a key role in tissue remodeling and regeneration [21]. However, most of the 3D cell-laden constructs are generally devoid of any exposure to fluid mechanical cues, such as fluid shear stress, tension, and compression in the maturation process. One potential approach to artificially generating the chemical and mechanical demands of human tissues is using complexly advanced in vitro culture systems, such as bioreactors [22]. A bioreactor is described as a simulator, which can be modified and controlled, including pH, temperature, oxygen tension, and perfusion of the cells, as well as external stimuli such as shear stress and mechanical forces [23]. Bioreactors have the ability to aid in the development of engineering human tissues from in vitro maturation of 3D cell-laden scaffolds after bioprinting.

In this review, we describe recent technological advances in 3D bioprinting for human tissues. The processes are beginning from data processing to bioprinting techniques, including inkjet, laser and extrusion-based bioprinting. We then review the commonly bioink formulation, including inks and cell selection, and key bioink properties for the bioprinting process. Various bioreactors that have been used for the delivery of biochemical and mechanical cues are discussed. The in vitro bioreactor systems used for the maturation of human tissues are summarized. Present limitations and future developments on the application of 3D bioprinting and bioreactor systems for engineering human tissues have been presented in the end.

## 2. Technique Approaches

### 2.1. Data Processing

3D bioprinting starts with a computer-assisted process to design a defined 3D biological model. The creation of a 3D model allows using data generated by computer-assisted design software (CAD, Solidwork) [24] or import data from medical imaging such as MRI and CT [15]. 3D model data from software design allows greater freedom of design, such as lattice and circle models. The generation of a 3D model using patient-specific tissue size and morphology enables the generation of a customized 3D construct that is a closer mimic of the human tissues. The 3D model is converted to a standard tessellation language (STL) file and then saved as a file format such as g-code. The file format could be easily followed by the printer and direct the layer-by-layer depositions of the biological elements [25].

### 2.2. Bioprinting Techniques

Several additive manufacturing techniques have been used to fabricate 3D scaffolds [26]. Among them, inkjet-based, laser-assisted, and extrusion-based bioprinting are popularly used to fabricate 3D cell-laden scaffolds for human tissues. Each bioprinting technique has specific strengths, weaknesses and limitations. Table 1 provides a concise comparison of these three approaches.

#### 2.2.1. Inkjet-Based Bioprinting

Inkjet bioprinting was the first bioprinting technology published in 2003 [40] and is very similar to conventional 2D inkjet printing [41]. Inkjet bioprinters deliver a controlled amount of bioink to the desired printing surface, forcing the content to flow continuously (continuous inkjet printing) or drop out from the nozzle (on-demand inkjet printing). The cell-laden biological solution is stored in the ink cartridge, and the electronically controlled elevator stage provides the control of the *Z*-axis in the inkjet printer. During bioprinting, the printer head was used as a thermal or piezoelectric actuator to generate droplets onto a substrate, which can support or form part of the final cell-laden construct, as shown in Table 1. The advantages of inkjet bioprinting include high-throughput capability, high resolution, inexpensiveness, reproducibility, and relatively high cell viability (>90%) [28,42,43,44]. Furthermore, this technique provides a useful method for depositing multiple cells [29,45] or proteins [31] onto a targeted spatial position with multiple print heads, therefore allowing the fabrication of complex multicellular constructs. Atala et al. [46] were fabricated human amniotic fluid-derived stem (AFS) cells-laden alginate/collagen scaffolds by thermal inkjet printing. The printed cell-laden constructs were cultured in the osteogenic inductive medium for 45 days and resulted in intensely mineralized nodules. Gao et al. [47] fabricated a bone-like tissue by delivering human mesenchymal stem cells (hMSCs) and hydroxyapatite or bioactive glass nanoparticles in strong poly(ethylene glycol) gel using a modified Hewlett-Packard (HP) inkjet printer. Results not only showed high cell viability post-printing but also hydroxyapatite enhanced the osteogenesis of the bioprinted hMSCs. However, one major drawback of inkjet bioprinting is the material of choice. The bioink must be in a liquid state and with appropriate viscosity to be ejected out of the small orifice of the nozzle [48].

#### 2.2.2. Laser-Assisted Bioprinting

Laser-assisted bioprinting originated from laser direct-write technology [49] and is a modified version of the laser-induced forward transform technique, which was developed to transfer biological material, such as peptides, DNA, and cells [33,35,50]. A typical laser-assisted bioprinter consists of five elements: (1) pulsed laser beam, (2) a focusing system, (3) a ‘ribbon’ structure donor layer containing an energy-absorbing layer that responds to laser stimulation, (4) a layer of liquid bioink solution, and (5) a receiving substrate for patterning and crosslinking bioink (Table 1). During the laser-assisted bioprinting process, the absence of direct contact between the bioink and the dispenser prevents cell stress resulting in high cell viability (>95%) [16]. Besides, laser-assisted bioprinting is compatible with different bioink types and a wide range of viscosities (1–300 mPa/s) [12]. Laser-assisted bioprinting is thus a promising technique for 3D printing of cell-laden constructs for human tissues. Catros et al. demonstrated that laser-assisted 3D bioprinting supported the fabrication of nano-hydroxyapatite (nHA) and human osteoprogenitor cells (HOP) without altering the physicochemical properties of nHA while keeping the viability, proliferation, and phenotype of HOPs [34]. Laser-assisted bioprinting has also been used for the deposition of a high concentration of human osteosarcoma cells (MG63) on 3D polycaprolactone (PCL) electrospun scaffolds with high cell viability in vitro and cell proliferation in vivo [51]. Furthermore, in vivo, laser-assisted bioprinting was used to deposit cell-laden nano-hydroxyapatite in a mouse calvaria 3D defect model. The preliminary results demonstrated that the 3D cell-laden constructs are possible to fabricated by laser-assisted bioprinting in in vivo [52]. Although there are many advantages to laser-assisted bioprinting, the influence of laser exposure on cells is not fully investigated. Additionally, the high equipment cost and the complexity of the laser printing control system represent another limitation for the use of this technique [16].

#### 2.2.3. Extrusion-Based Bioprinting

The extrusion-based bioprinting technique combines both a fluid-dispensing system and an automated robotic system for extrusion and printing, respectively [53]. The fluid-dispensing system can be driven by a pneumatic- or mechanical-based system as a “power source”. By applying a continuous force during the bioprinting process, bioink is printed in uninterrupted cylindrical lines rather than a single bioink droplet. Under the control of the automated robotic system, the cylindrical filaments can be precisely fabricated to the desired 3D custom-shaped structures (Table 1). Recently, the development trend of an extrusion-based bioprinter is multi-head tissue construction. A typical extrusion printer can be seen in the Dong-Woo Cho’s group, which has a three-axis motion control with six dispensing heads, supporting up to six different bioinks printing [38]. 

One of the main advantages of extrusion bioprinters is their ability to print a wider range of biomaterials with varying viscosity including hydrogels, biocompatible copolymers, and cell spheroids from 30 to 6 × 10^7^ mPa/s [54]. Normally, the higher viscosity materials often provide structural support for the printed construct and lower viscosity materials provide a suitable environment for maintaining cell viability and function. One of the disadvantages is the encapsulated cells are exposed to larger stresses reducing cell viability during bioprinting [12]. Several researchers have used extrusion-based methods for bioprinting of human tissues [24,55,56,57,58]. Kang et al. have fabricated stable, human-scale mandible, and calvarial bone tissue construct using an integrated tissue-organ printer (ITOP). This system was based on extrusion-based bioprinting and 3D human amniotic fluid-derived stem cells (hAFSC)-laden hydrogels composing scaffolds were fabricated. In vivo results showed newly formed vascularized bone tissue throughout the implants [15]. Wüst et al. showed that hMSCs encapsulated in alginate and hydroxyapatite constructs were not damaged during the printing process and the cells showed high viability in in vitro culture [24]. Fedorovich et al. developed heterogeneous bone constructs containing endothelial progenitor cells (EPC) and MSCs to promote neovascularization during bone regeneration. Following in vivo implantation, the EPCs assembled more blood vessel networks than MSCs during bone formation [55].

## 3. Bioink Formulation and Key Bioink Properties

### 3.1. Inks

The scaffolding material used for bioprinting is called bioink. Bioink consists of a biomaterial solution (ink) and cells in the presence or absence of growth factors. Bioink formulation is one of the main challenges in the 3D bioprinting of cell-laden scaffolds for human tissues. One of the reasons for that is the physical and chemical cues of the cell hosting biomaterials requires an understanding of cell physiology and cell-ECM interaction. Natural (gelatin, collagen, fibronectin, alginate, chitosan, silk fibroin, and hyaluronic acid) or synthetic polymer (polyethylene glycol (PEG), PCL, poly (lactic-co-glycolic) (PLGA), polylactide (PLA)) were used for tissue engineering. The advantages of natural polymers for tissue engineering applications are their similarity to human ECM and their inherent bioactivity. Synthetic polymers can be tailored with specific physical properties to suit particular applications. To combine the advantages of natural and synthetic polymers, some bioinks are hybrid biomaterials that merge natural and synthetic materials.

Decellularized extracellular matrices (dECM) have been an increasingly promising material in tissue engineering. Hydrogels made from decellularized tissues including urinary bladder, heart, liver, dermis, adipose tissue, bone, and lung, among others, were developed and reported to support growth and function of different cell types. Pati et al. showed that dECMs from three tissues (cartilage, heart, adipose tissue) could be solubilized into bioinks and subsequently be bioprinted [59]. dECM bioinks contain the diverse array of ECM components characteristic of different tissues and, as a result, more closely resemble the native tissue. Although the low viscosity of dECM bioinks compromise mechanical properties and shape fidelity of the bioprinted 3D construct, they represent a promising addition to the bioinks.

A summary of recent outstanding bioprinting studies for tissue engineering is shown in Table 2, including bioprinting technique, materials, concentration, cell type, cytocompatibility, and applications. According to the wide variety of hydrogels that have been bioprinted, there are three different crosslinking mechanisms: chemical (ion compound [60], pH [61]), physical (temperature [62], light [63]), and enzymatic [64] crosslinking. The gelation processes of bioink will sometimes include several crosslinking mechanisms to print stable and complex scaffolds [65,66].

### 3.2. Cell Selection

Tissue regeneration is known to be a well-orchestrated process in which stem cells play a major role together with growth factors [93]. Stem cells are characterized by their ability to self-renew and differentiate into a variety of functional specialized cell types. They are several stem cell types for engineering human tissues used in 3D bioprinting processes, such as hMSCs [24,67,71,87,88,89,94], adipose-derived stem cells (ASC) [95], hTMSCs [59,92], and human amniotic fluid-derived stem cells (hAFSC) [15,59] (Table 2). In most cases, it is necessary to add external supplements in the culture medium to induce differentiation of the stem cells to a targeted cell phenotype. For example, in vitro differentiation of stem cells into the osteoblasts and osteocytes lineage require to supplement the cells’ culture medium with specific compounds called osteogenic medium (including β-glycerophosphate, ascorbic acid, and dexamethasone). Riccardo et al. have shown that the 3D bioprinted MSCs-laden GelMA/gellan gum scaffolds support the osteogenic differentiation of MSCs and bone matrix deposition when cultured in the osteogenic media [87]. Another cell type in bioprinting strategies is the usage of tissue-specific cell types, such as preosteoblasts and osteoblasts for bone tissue [34,73,74,75,79,80,94], chondrocytes for cartilage tissue [66] and adipocytes for adipose tissue [96]. Those cells have a stable tissue phenotype and have been used to regenerate tissues. Neufurth et al. have printed alginate/gelatin/human osteoblast-like SaOS-2 cells scaffolds, consisting of agarose and the calcium salt of polyphosphate, which resulted in a highly significant increase in cell proliferation and mineralization [80].

Meanwhile, to mimic the human tissues does not only require engineered complex constructs but also represents the cell type diversity of the tissue. The ability of 3D bioprinting techniques to simultaneously print different cell types with spatial accuracy has garnered much interest. Co-culture of chondrocytes and MSCs or MG63 cells in hydrogels have been investigated for improved chondrogenesis and osteogenesis [56,97]. Shim et al. utilized a multi-head tissue building system to separately dispense human chondrocytes and MG63 cells, which were used to fabricate osteochondral tissue [97]. Human umbilical vein endothelial cells (HUVEC) or endothelial progenitor cells (EPC) have been shown to produce blood vessels, especially when seeded with osteogenic cells or MSCs. The combination of HUVECs or EPCs with osteogenic cells has been studied for angiogenesis and osteogenesis [55,98]. Fedorovich et al. have shown bone formation in constructs that contain MSCs and EPCs. EPCs derived from peripheral blood contribute to osteogenic differentiation of MSCs in vitro, and MSCs support the proliferation of EPCs and stabilize the formed cellular networks. After in vivo implanted in the mice separate subcutaneous dorsal model, EPCs from peripheral blood assembled into early blood vessel networks [55].

### 3.3. Growth Factor Selection

Growth factors are soluble signaling molecules that control a wide variety of cellular responses, such as cell growth, proliferation, and differentiation through specific binding of transmembrane receptors on target cells [99]. The idea to use growth factors to promote tissue regeneration is intuitive, as growth factors are highly related to the repair of damaged human tissues. For example, transforming growth factor-β (TGF-β) [100,101], insulin-like growth factors (IGF) [102], bone morphogenetic proteins (BMP) [103], vascular endothelial growth factor (VEGF) [104], and parathyroid hormone (PTH) [105] are among the most extensively used growth factors and hormones to stimulate the differentiation of stem cells (Table 3). TGF-β superfamily plays an important role in embryonic development, tissue morphogenesis, cell proliferation and cell differentiation. TGF-β1 and TGF-β3 have been used for chondrogenic differentiation and chondrogenic phenotype maintenance of MSCs for cartilage and osteochondral tissue regeneration. However, TGF-β has only provided limited success for endochondral bone formation in adult non-human primates [106]. BMPs, particularly BMP-2, BMP-4, and BMP-7, are the most extensively studied osteogenic molecules for inducing de novo bone formation in ectopic and orthotropic sites, including critical size defects [107]. VEGF and IGF can regulate angiogenesis, and bone research with angiogenic factors has primarily focused on VEGF’s role in neovascularization and osteogenic recruitment [108]. Although PTH mechanisms for directing osteogenic activity are not well understood, studies have shown that periodic exposure of PTH can stimulate bone formation in rats and humans [109].

Among all the delivery methods, such as freedom adding in culture medium and microsphere delivery, 3D bioprinting provides a promising approach to incorporate growth factors into hydrogel scaffolds more easily compared to in a spatiotemporal distribution. Du et al. created a collagen-binding domain (CBD), which collagen microfibers bound BMP-2. The results show that BMP-2 was able to be controllably released in vitro. The CBD-BMP2-collagen microfibers induced the differentiation of MSCs into osteocytes within 14 days more efficiently than the osteogenic media [88]. Spatial patterns of BMP-2 with a 10 mg/mL concentration were printed on a fibrin-coated glass surface using an inkjet bioprinter. Murine muscle-derived stem cells seeded onto the BMP-2 pattern exhibited alkaline phosphatase (ALP) activity, indicating osteogenic differentiation [110].

### 3.4. Key Bioink Properties for 3D Bioprinting of Human Tissues

During the selection of foundation components (hydrogel, cell, growth factor), we need to consider the bioink’s printability. The suitability of a bioink for the bioprinting process mainly depends on its physicochemical properties under the conditionals imparted by the specific bioprinting parameters. One of the specific bioprinting parameters is nozzle gauge, which will consequently determine the resolution of the scaffold, fabrication speed, and time, as well as the shear stress at which embedded cells are exposed to during the printing process. The major physicochemical properties that determine the printability of a bioink are its rheological properties and crosslinking mechanisms. In the rheological parameters, viscosity is the resistance of a fluid to flow upon the application of stress. The polymer type, concentration, and molecular weight determine the viscosity of a polymer solution. Printing fidelity generally increases with increasing viscosity [111]. However, an increase in viscosity implies an increase in applied shear stress, which may be harmful to the suspended cells [112]. Gelation of crosslinking of a printed bioink structure is necessary to preserve its 3D construct with structural integrity. The crosslinking mechanisms are determined by the hydrogels chosen for printing, and normally it can be either physical or chemical or a combination of both mechanisms. Physical crosslinking mechanisms rely on non-chemical interactions, including ionic [74,94], stereo complex [113], and thermal crosslinking [114]. Physically crosslinked hydrogels are the most prominent hydrogel class used for bioprinting, but a significant drawback is their poor mechanical properties. Chemical crosslinking forms newly covalent bonds to connect gel precursors. Chemical crosslinking may provide the hydrogel with good handling properties and high mechanical strength but needs a very stringent control of crosslinking kinetics. The readers are referred to the paper by Malda et al. [27] for detailed information about how rheological properties and crosslinking mechanisms affect 3D bioprinting processes and structure fidelity. Except for the printability, the ink features, such as biocompatibility, biodegradability, mechanical property, and material biomimicry, were important for scaffold maturation to achieve the functional human tissues (Table 4).

## 4. In Vitro Bioreactor Systems for Scaffold Maturation

After bioprinting, static cultivation is the main culturing approach for scaffold maturation. 3D bioprinted cell-laden scaffolds are statically cultivated in the incubators, covered with media that has to be exchanged manually. Several disadvantages follow the static cultivation, like mass-transfer limitations of nutrients and oxygen or waste removal. Different types of bioreactors have been designed for scaffold maturation allowing dynamic cultivations adapted to the requirements of individual cells or tissues. The dynamic bioreactor system enables the control and monitoring of pH value, O_2_ saturation, flow rate, and temperature, as well as mechanical stimulation.

The choice of a bioreactor to cultivate 3D cell-laden scaffolds after bioprinting depends upon the tissue to be engineered and its functional biomechanical environment. For example, Wolf’s law indicates that bone strength increases and decreases as the mechanical forces on the bone increase and decrease [115]. Mechanical loading is applied to the bone causes fluid to flow through the lacuna-canalicular system in bone, and osteocytes sense to the shear stress and induce osteoblasts to form bone or osteoclasts to resorb bone [116]. Therefore, bioreactors designed for bone tissue, compression, shear stress and perfusion are constantly highlighted. Emulation of physiological conditions has been addressed in different ways and the incorporation of convective forces has become a common characteristic among most bioreactors. In this section, we describe some of the most common bioreactors found in the engineering of several functional tissues such as bone, cartilage, skin, and kidney applications.

### 4.1. Spinner Flask Bioreactor

The spinner flask was first designed with the idea to use convection in order to maintain a well-mixed system. It consists of a dual-side arm cylindrical flask with a rubber stopper serving as a cover, which is shown in Figure 2A. 3D scaffolds are threaded into needles connected to the cover of the flask and submerged in the culture medium. A magnetic stir bar or a shaft is used to generate cell culture media convection, which provides a homogeneous distribution of oxygen and nutrients surrounding the scaffolds [117]. Spinner flasks bioreactors have been shown as an effective method for large-scale in vitro chondrogenic differentiation and the subsequent in vivo cartilage formation of human adipose-derived stem cells (ADSCs) [118]. Mygind et al. [119] have found that dynamic spinner flask cultivation of hMSCs-laden hydroxyapatite scaffold constructs resulted in increased proliferation, differentiation, and distribution of cells in scaffolds compared to static controls. Stiehler et al. [120] worked on the same dynamic spinner flask system and repeated the experiment on PLGA scaffolds for up to 3 weeks. They demonstrated a 20% increase in DNA content (day 21), enhanced ALP specific activity (7 days and 21 days), a more than tenfold higher Ca^2+^ content (21 days), and significantly increased transcript levels of early osteogenesis markers (e.g., COL1A1, BMP2, RUNX2) in spinner flask culture compared to static culture. However, the reasons for their success in the spinning flask culturing remain inconclusive and the mechanical stimuli induced by the magnetic stir bar or the shaft may contribute to the functional tissue formation. Melke et al. [121] have demonstrated that the complex flow within the spinner flask and mechanical stimulation on the scaffold were different when culturing at 60 and 300 RPM. Results show that culturing at 300 RPM led to a more homogeneously distributed ECM than at 60 RPM, which is mainly at the bottom of the scaffold. Those results were in agreement with the computational simulations that predicted maximal scaffold mineralization based on different wall shear stress stimulation. Despite these advantages, the disadvantages of spinner flask cultures showed that the magnitude of the shear stresses could vary significantly between different locations; therefore, not all the cells are exposed to the same shear stresses.

### 4.2. Rotating-Wall Vessel Bioreactor

The rotating-wall bioreactor consists of two concentric cylinders whose annular space contains the cell culture medium, which is shown in Figure 2B. The inner cylinder is static and permeable to allow CO_2_ gas exchange for oxygen supply. The outer cylinder is impermeable and horizontally rotates at a speed that causes centrifugal forces that can balance the gravitational forces. Continuous rotation of the outer cylinder results in the gentle falling of cells through the medium while remaining in suspension. The rotating-wall vessel bioreactor is an optimized suspension culture system in which cells are grown in a physiological low fluid shear environment in 3D. To date, more than 50 rotating-wall vessel-derived tissue models have been engineered, including bone, cartilage, liver, neuronal tissue, cardiac muscle, adipose tissue, and epithelial tissues [126,127,128,129,130]. Song et al. [131] demonstrated that rat osteoblasts cultured in rotating wall vessel bioreactors expanded by more than 10 times compared to osteoblasts in spinner flasks and static controls, and they presented better morphology, viability, and stronger ability to form bone tissue. Human cartilage progenitor cells have also been shown to differentiate into mature chondrocytes using a combination of scaffold and rotating-wall vessel cultivation [132]. Cardiac tissue has also been engineered using rotating-wall vessel cultivation, which gave rise to a highly differentiated tissue that exhibited normal anisotropic electrophysiological properties [126]. However, the transport of nutrients to the center of the scaffold was still limited because the convective forces could not extend to the interior of the large-scale constructs. Large rotation speeds of the outer wall will increase mass transport, whereas an increase in the differential rotation enhances shear stresses on scaffolds [133].

### 4.3. Compression Bioreactor

Compression bioreactors are made of a compression chamber with one or more pistons, which applies compressive loads directly to the scaffolds [123,124,134] (Figure 2C). Generally, the supporting facilities such as the mechanical stimulation unit allow control on the loading frequency, strain, force, and time. Compression bioreactors are intended to mimic the natural physiological loading of tissues in vivo and they are becoming more widely used in bone and cartilage tissue engineering. Mauck et al. [135] demonstrated that the application of dynamic deformational loading at physiological strain levels enhances chondrocyte matrix elaboration in cell-seeded agarose scaffolds to produce a more functional engineered cartilage tissue construct than in free swelling controls. Compression bioreactors can improve glycosaminoglycan and hydroxyproline formation and increase the elastic modulus of the cartilage formed to approach that of native cartilage [136,137]. Meanwhile, Sittichockechaiwut et al. [138] have shown that osteoblasts were highly sensitive to mechanical loading. The compression loading at 1 Hz, 5% strain has a strong effect on mineralized matrix production and osteogenic-related gene expression in comparison to static conditions. Compression bioreactor design and mechanical loading protocols are varied in different tissues to be engineered.

### 4.4. Perfusion Bioreactor

Flow perfusion bioreactors provide continuous culture medium flow through scaffold, which generates shear stress on cells. The culture medium is continually recirculated through the chamber by directly pumping, thus improving the transport of nutrients and oxygen to the constructs. Perfusion bioreactors are useful in large tissue mass constructs as it allows for more precise control of the culturing environment. Meanwhile, structural parameters of the scaffold-like porosity or permeability have a significant influence on the experimental outcome in perfusion cultures. Vetsch et al. [125,139] have developed a perfusion bioreactor system (Figure 2D) to produce shear stress forces on the engineered bone-like tissues scaffolds. The designed bioreactors enable to non-invasively and temporally monitor the development of mineralized ECM by the monitoring of micro-CT. Vetsch et al. [139] have investigated the influence of curvature on three-dimensional mineralized matrix formation under static and perfused conditions. The results showed that the ingrowth of mineralized tissue into the channels was dependent on curvature and was higher under perfusion. Large channels were not closed in any group compared with partially (static) or fully (perfused) closed medium and small channels. Mineralized tissue morphology was cortical-like in static samples and trabecular-like in perfused samples. The flow rate in the perfusion bioreactor system is one of the most essential parameters for tissue engineering. The optimal range varies according to the design of the bioreactor and the cell type used. For instance, increasing flow rates of 0.075–0.2 mL/min to human chondrocytes seeded on PLGA scaffolds for up to 5 weeks increased the percentage of glycosaminoglycan retained in the ECM [140]. Vetsch et al. [125] showed that mineralized extracellular matrix formation was completely inhibited at a low flow rate (0.001 m/s) compared to a high flow rate (0.061 m/s) and the static group. Biochemical assays and histology confirmed these results and showed enhanced osteogenic differentiation in the high flow rate group. Meanwhile, Zhao et al. [141] demonstrated that the various ranges of optimal flow rates to induce mineralization are within 0.5–5 mL/min among different hMSCs-laden silk fibroins scaffolds in the perfusion bioreactor by combining computational fluid dynamics and mechano-regulation theory.

## 5. Present Limitations and Future Perspectives

The challenges facing the 3D bioprinting of cell-laden scaffolds for tissue regeneration field relate to specific technical, material and cellular aspects of the bioprinting process. Although 3D bioprinting techniques offer a precise and structured approach for tissue engineering, there are some significant challenges to create tissue constructs of clinical relevance. For example, bone is a metabolically active tissue supplied by an intraosseous vasculature [142]. Angiogenesis occurs spontaneously upon implantation of a bone graft, but host-derived neovascularization of the implant is slow (<1 mm/day) [37], and thus insufficient for constructs. As complex engineered 3D constructs of clinically relevant size cannot be sustained by the diffusion of nutrients alone, creating a functional vascular network is necessary for ensuring nutrient supply and waste removal [37]. Approaches to potentially improve vascularization bioprinting include computer simulation, microscale technology, and sacrificial printing. Computer modeling is a powerful tool in designing engineered tissue with the desired properties, such as gradient porosity and mechanical properties [143,144]. The use of computer-simulated models to optimize the design of the vasculature network will be an empowering tool to increase nutrient and waste efficiency. Microscale technology offers flexibility in creating precise 3D architectures with embedded vascularized and capillary networks, with a layer-by-layer assembly. This approach creates a trench that is molded into one layer before a second layer is aligned and deposited, forming laminated channels or grooves in an iterative manner [145] (Figure 3A). Although promising, this method is slow. Small bioprinted tissue may take only minutes or hours to print, but the question of cell viability both within a pre-polymer bioink and within the polymerized early regions of a large multi-day print must be addressed. Another alternative strategy has been generated a vascular network using 3D printing sacrificial biomaterials, such as gelatin, Pluronic F-127 (Figure 3B), and carbohydrate glass [145]. The sacrificial materials print vascularization channels and provide mechanical support at each layer during fabrication, and then are removed from the completed object in a post-processing step. This method will increase the complexity of the printing process and the method of removal and breakdown products must be cytocompatible. Although all these methods are not reliable approaches to print pre-vascularized tissues, a faster bioprinter with higher resolution would be poised to solve some of the problems.

The ink selection remains a major concern and limitation in 3D bioprinting cell-laden scaffolds for tissue regeneration, as the selected materials should consider both their compatibility with cell growth and functions and their printability characteristics. For this reason, many published studies select a limited range of materials, including alginate, gelatin, collagen, silk fibroin, chitosan, PEG and agarose. Meanwhile, each type of biomaterial has specific advantages and disadvantages. The common approach is multi-material printing. It can not only better mimic native organic and inorganic hybrid components of bone tissue, but also provide a way to improve shape fidelity and mechanical strength. Kang et al. have fabricated a human-sized structural integrity calvarial scaffold with printing hAFSCs-laden composite hydrogel with poly(ε-caprolactone) (PCL)/tricalcium phosphate (TCP) framework using an integrated tissue-organ printer. The results showed large blood vessel formation within newly formed bone tissue throughout the bioprinted bone constructs, including the central portion [15]. Incorporating multiple materials also remains a challenge in creating gradients of cells or growth factors due to the need to prepare many independent solutions [16]. Further smart biomaterials need to be developed. Promising developments are the generation of self-assembly materials and stimulus-responsive hydrogels. Self-assembly is the way to originate materials (nanoparticle or hydrogel), cells, and proteins to produce novel supramolecular architecture at micro-levels, which will provide a way to produce complex combinations and gradients of native bone ECM components [146]. Stimulus responsive hydrogels can be classified into mechano-, chemo-, heat, pH, and light-responsive hydrogels. Bioprinted constructs with shapeshifting ability can be formed through placing hydrogels with different stimulus responses strategically [147]. Moreover, the degradation of hydrogel scaffolds can be tailed through incorporating cell-responsive sites.

In the end, different kinds of forces, such as shear stress and compression loading in bone tissue, have a synergism effect in the native tissue development and remodeling processes. To mimic the natural microenvironment of tissue, bioreactors are developed to apply combined mechanical force on 3D cell-laden scaffolds. Shahin et al. [149] illustrated that human chondrocytes benefit from the combined application of intermittent unconfined shear and compressive loading at a frequency of 0.05 Hz using a peak-to-peak compressive strain amplitude of 2.2% superimposed on a static axial compressive strain of 6.5% for 2.5 weeks. Glycosaminoglycan and collagen type II productions were enhanced between 5.3- and 10-fold after simultaneous stimulation. We foresee that future research would be centered on a more complex mechanic system that mimics the in vivo mechanical loading condition of natural tissue. Figure 3E,F shows a future trend of 3D bioprinting multiple cells-laden scaffolds for cardiac tissue engineering combined with complex bioreactors with different stimulations, such as biochemical, mechanical, electric, and perfusion. After in vitro scaffold maturation in a bioreactor system, a whole functional heart is formed.

## 6. Conclusions

In this review, we focus on the 3D bioprinting of cell-laden scaffolds for human tissue engineering applications. Due to advantages in micro-scale, high-throughput, and cell deposition, bioprinting has become a strong fabrication tool to create complex micro-and macro-scale biomedical systems. The recent advances in different 3D bioprinting techniques, bioink consideration for 3D bioprinting, including hydrogel, cells, growth factors selection, and ink properties were systematically summarized. Advanced bioreactor systems providing the dynamic cultivation and mechanical stimulation to mimic the native human tissues have promising applications for scaffold maturation in vitro. Limitations of the technology and outlines promising directions for future prospects are further addressed. Overall, 3D bioprinting is an advanced fabrication technique for the fabrication of 3D cell-laden constructs for human tissue, with a bright future but encompassing numerous challenges and problems.

## Figures and Tables

**Figure 1 ijms-22-03971-f001:**
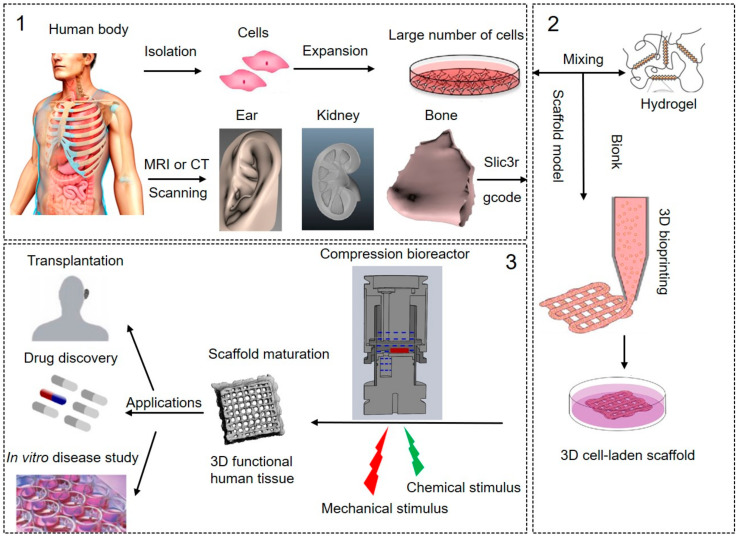
The processes of 3D bioprinting of human tissues. (**1**) In pre-processing: isolation of cells from the human body and in vitro cell expansion, Magnetic resonance imaging (MRI) or Computed tomography (CT) scanning were used to achieve the structure information of the target tissue and create the printing model, such as ear, kidney, and bone; (**2**) In processing: bioink preparation, 3D bioprinting of 3D cell-laden scaffolds guided by the MRI or CT scanning tissue models; (**3**) In post-processing: bioreactor culture system for in vitro scaffold maturation to be 3D functional human tissues, and potential applications of the 3D bioprinted human tissues. Pictures modified with permission from Reference [16].

**Figure 2 ijms-22-03971-f002:**
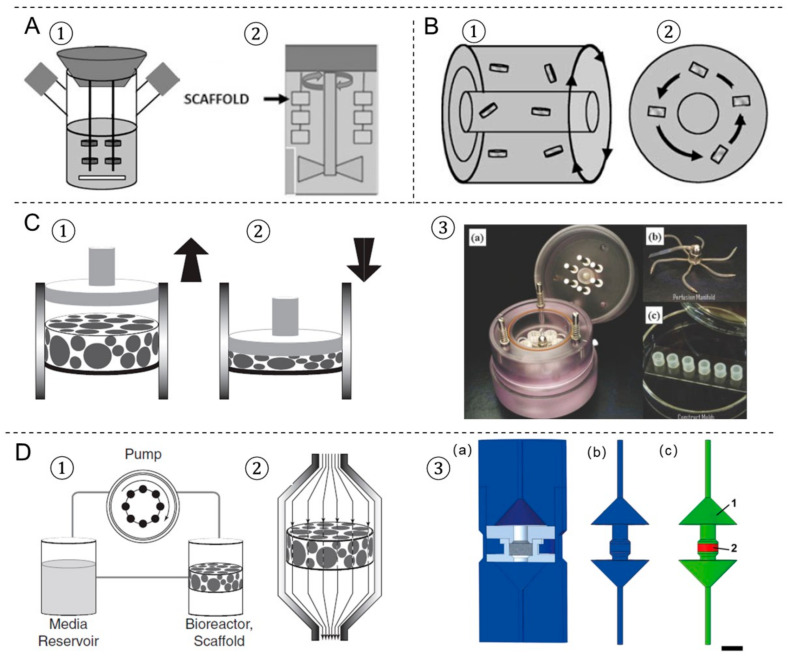
The four most prevalent bioreactors used in tissue engineering. (**A**) Spinner flask bioreactor using a magnetic stir bar (**1**) or a shaft (**2**). (**B**) Rotating wall vessel bioreactor in different views (**1**,**2**). (**C**) Compression bioreactor: a piston applies a direct dynamic compression load on the scaffold construct, (**1**) release, (**2**) load. (**3**) A bioreactor system of dynamic mechanical stimulation for engineered cartilage constructs, (**a**) at the bioreactor base is a culture medium reservoir, (**b**) the lid of the bioreactor, (**c**) the engineered constructs are molded into cylindrical plugs. (**D**) Perfusion bioreactor: (**1**) the perfusion system consists of a pump, a media reservoir and the bioreactor housing the scaffold; (**2**) the scaffold is press-fitted into a perfusion chamber to ensure medium flow through the scaffold. (**3**) Schematic bioreactor drawings of an in-house designed perfusion bioreactor for engineered bone tissue. (**a**) 3D computer-aided design model, (**b**) inverted volume of perfusion bioreactor, (**c**) material definitions for computational fluid dynamics model. Picture modified with permission from reference [122,123,124,125].

**Figure 3 ijms-22-03971-f003:**
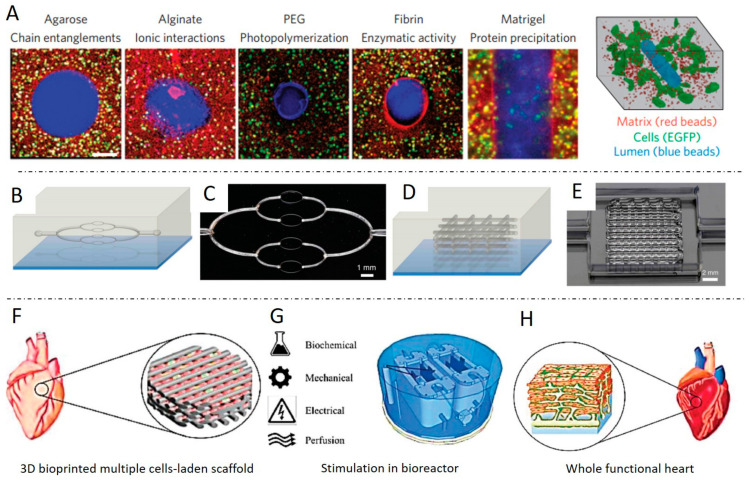
Examples of future perspectives of 3D bioprinting cell-laden scaffolds for tissue regeneration. (**A**) Carbohydrate glass to cast vascular features into a variety of hydrogels, forming perusable vessels that support cell growth. Schematic illustrations and optical images, of 2D (**B**,**C**), and 3D (**D**,**E**) embedded vascular networks that are printed, evacuated, and perfused with a water-soluble fluorescent dye. (**F**–**H**) A whole functional heart is formed from 3D bioprinted multiple cells-laden scaffolds cultured in the complex bioreactor combined with different stimulations. Pictures modified with permission from references [39,122,139,145,148].

**Table 1 ijms-22-03971-t001:** Comparison of three types of bioprinting techniques. Figures adapted with permission from [27].

	Inkjet-Based Bioprinter	Laser-Assisted Bioprinter	Extrusion-Based Bioprinter
Bioprinters	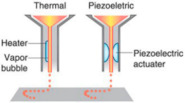	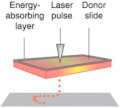	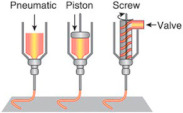
Cost	Low	High	Moderate
Material viscosities	3.5–12 mPa/s	1–300mPa/s	30 to >6 × 10^7^ mPa/s
Resolution	10–100 µm	~75 µm	100 µm-mm range
Quality of structure	Poor	Fair	High
Print speed	Fast	Medium	Slow
Cell viability	>85%	>95%	40–80%
Examples	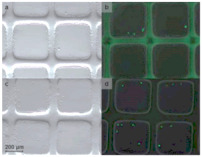	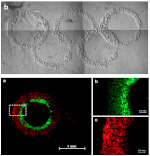	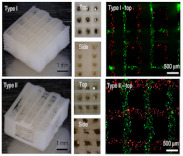
Reference	[25,28,29,30,31]	[32,33,34,35]	[15,24,36,37,38,39]

**Table 2 ijms-22-03971-t002:** A summary of outstanding recent bioprinting studies for human tissue.

Bioprinter Types	Material	Polymer Concentration (*w*/*v*)	Cell Type	Cytocompatibility	Application	Reference
Inkjet-based	alginate/collagen	na/0.1%	hAFSC	na	bone, brain	[46]
		1%/0.3%	hAFSC dSMC bEC	90%, day 7	vascular, bone	[11]
	PEGDMA/HA	20%/2%	hMSC	86%, day 21	bone	[47]
	PEGDMA/ GelMA	10%/1.5%	hMSC	80%	bone, cartilage	[67]
Laser-assisted	HA	60%	MG63	~100%, day 2	bone	[68]
	alginate/HA	0.5%/15%	HOP	high, day 15	bone	[34]
	alginate	1%	MG63	high, day 4	bone	[51]
	alginate	2%	hMSC	na	bone, cartilage	[69]
	fibrinogen/hyaluronic acid	1.3%/1%	ECFC	98%, day 0	adipose, vascular	[70]
Extrusion-based	alginate	3%	MG63	80%, day 0	TE (general)	[68]
		3.5%	MSCs	93%, 4 h	bone	[71]
		3%	MC3T3	93%, day 1	bone, liver	[72]
		3.5%	MC3T3	94%, day 1	bone	[73]
		3.5%	MC3T3	85%, day 1	bone, vascular	[74]
		3%	MG63	95%, 4 h	bone, TE	[75]
		10%	hMSC	85%, day 7	bone	[58]
		10%	MSC	89%, 5 h	bone	[56]
	alginate/gelatin	0.8%/4.1%	hMSC	85%, day 14	bone	[76,77]
	alginate/gelatin/graphene oxide	0.8%/4.1%/0.1%	hMSC	92%, day 42	bone	[78]
	alginate/collagen I/GAG	4%/2%/na	MC3T3	88%, day 1	bone, liver	[79]
	alginate/gelatin	5%/5%	SaOS-2	92%, day 1	bone	[80]
	alginate/gelatin/HA	2%/10%/8%	hMSC	85%, day 3	bone	[24]
	alginate/gelatin/carboxymethyl chitosan	1%/10%/0.1%	hMSC	>85%, day 2	TE (general)	[81]
	alginate/GelMA	4%/4.5%	HUVEC	80%, day 1	TE (general)	[82]
	alginate/matrigel/CaP	10%/na/10%	MSC/EPC	na	bone	[83]
	alginate/PCL	3.5%/na	MSC	na	bone	[84]
	agarose	1.5%	hMSC/ MG63	excellent, day 21	TE (general)	[85]
	collagen I	2%	MG63	>90%, day 14	bone, cartilage	[86]
	GelMA/gellan	10%/1%	MSC	90%, day 3	bone	[87]
	GelMA/collagen	5–20%/0.01–0.1%	MSC	95%, day 28	bone	[88]
	gelatin/fibrinogen/hyaluronic acid	3.5%/2%/0.3%	hAFSC	91%, day 1	bone, ear, muscle	[15]
	matrigel/CaP	na/15%	MSC	81%, day 1	bone	[89]
	matrigel	na	MSC	86%, day 7	bone	[57]
	alginate	2%	MSC	>86%, day 14		
	F127	25%	MSC	4%, day 3		
	agarose	1%	MSC	70%, day 7		
	MeHA	2.5%	fibroblast	96%, day 0	TE (general)	[90]
	GelMA	5%	fibroblast	95%, day 7		
	PEGDA	5%	fibroblast	>87%, day 7		
	NorHA	2%	fibroblast	>87%, day 7		
	silk fibroin	3%	MC3T3	70%, day 2	TE (general)	[91]
	silk fibroin/gelatin	8%/15%	hTMSC	96%, day 1	bone, cartilage	[92]

HA: hydroxyapatite; PEGDMA: poly(ethylene glycol)dimethacrylate; GelMA: gelatin methacrylamide; GAG: gly-cosaminoglycan; MeHA: methacrylated hyaluronic acid; NorHA: norbornene-functionalized hyaluronic acid. hAFS: human amniotic fluid-derived stem cell; dSMC: canine smooth muscle cell; bEC: bovine aortic endothelial cell; MSC: mesenchymal stem cell; HOP: human osteoprogenitors; MG63: human osteosarcoma cell; SaOS-2: human osteogenic sarcoma cell; MC3T3: preosteoblast cell; ECFC: endothelial colony-forming cell; HUVEC: human umbilical vein endo-thelial cell. na: not available; TE: tissue engineering.

**Table 3 ijms-22-03971-t003:** A summary of growth factors in tissue regeneration. Adapted with permission from Reference [37].

Growth Factor	Tissues Treated	Representative Function
TGF-β3	Bone, cartilage	Proliferation and differentiation of bone-forming cells The antiproliferative factor for epithelial cells Enhances hyaline cartilage formation in vivo
IGF-1	Muscle, bone, cartilage	Cell proliferation and differentiation of osteoprogenitor cells, inhibition of cell apoptosis
BMP(-2, -7)	Bone, cartilage	Differentiation and migration of osteoblasts Enhanced bone healing and increased bone mechanical strength in the majority of patients
VEGF	Bone, blood vessel	Enhanced vasculogenesis and angiogenesis
FGF (-1, -2, -18)	Blood vessel, bone, muscle	Migration, proliferation and survival of endothelial cells, inhibition of differentiation of embryonic stem cells, increased osteogenic differentiation of mesenchymal stem cells
HGF	Bone, muscle	Proliferation, migration and differentiation of mesenchymal stem cells
PDGF-AB (or -BB)	Blood vessel, muscle, bone, cartilage	Proliferation, migration, growth of endothelial cells Osteoblast replication and collagen I synthesis in vitro
PTH, PTHrP	Bone	Accelerated bone healing through upregulation of bone markers and resultant bone tissue Bone formation was activated in postmenopausal females but inhibited in healthy adults

**Table 4 ijms-22-03971-t004:** Ink features for 3D bioprinting of human tissue. Table modified with permission from reference [12].

Printability	Physicochemical properties (surface tension, viscosity, crosslinking) of the ink that allows its spatial and temporal deposition with high precision and accuracy during the printing process.
Biocompatibility	The ability of the ink to support normal cellular activity (cell attachments and proliferation) without causing an inflammatory or immune response to the host tissue.
Biodegradability	The ideal degradation rate of ink is matching the ability of cells to replace the ink material with their extracellular matrix proteins. Degradation by-products should be harmless and easily metabolized from the host.
Mechanical property	Bioinks should provide the required tensile strength, stiffness, and elasticity for mimicking the mechanical properties of native bone tissues and provide the cells with a stable environment for attachment, proliferation, and differentiation.
Material biomimicry	Engineering bioink material with specific physiological functions requires mimicking the naturally tissue-specific composition and localization of extracellular matrix components in the human tissue.

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
