# Peer review of "3D Bioprinting of Human Tissues: Biofabrication, Bioinks, and Bioreactors"

_ijms, 2021, doi:10.3390/ijms22083971_

Round 1

Reviewer 1 Report

This manuscript, written by Zhang et al., focuses on recent technological advances in 3D bioprinting of cell-laden scaffolds for human tissue engineering applications. Authors herein review the major bioprinting techniques (inkjet-based bioprinting, laser-assisted bioprinting, extrusion-based bioprinting), bioink formulation (inks, cell selection, growth factor selection), key bioink properties for 3D bioprinting of human tissues, and in vitro bioreactor systems for scaffold maturation (spinner flask bioreactor, rotating-wall vessel bioreactor, compression bioreactor, perfusion bioreactor). They also provide a summary of present limitations and future developments on the application of 3D bioprinting and bioreactor systems for engineering human tissues. The manuscript, in general, is well-written. I suggest the acceptance of this manuscript in its present form.

Reviewer 2 Report

See attached file.
